# Impact of the Poor Oral Health Status of Children on Their Families: An Analytical Cross-Sectional Study

**DOI:** 10.3390/children8070586

**Published:** 2021-07-09

**Authors:** Mir Faeq Ali Quadri, Fatimah Rasheed M. Jaafari, Noha Ahmed A. Mathmi, Nouf Hassan F. Huraysi, Maryam Nayeem, Abbas Jessani, Santosh Kumar Tadakamadla, Jyothi Tadakamadla

**Affiliations:** 1Dental Public Health, Department of Preventive Dental Sciences, College of Dentistry, Jazan University, Jazan 45142, Saudi Arabia; 2Dental Intern Trainee, College of Dentistry, Jazan University, Jazan 45142, Saudi Arabia; fatimahRJ@hotmail.com (F.R.M.J.); Noha95@windowslive.com (N.A.A.M.); drnoee1995@gmail.com (N.H.F.H.); 3Department of Pharmacology, College of Pharmacy, Jazan University, Jazan 45142, Saudi Arabia; maryamnayeem12@gmail.com; 4Schulich School of Medicine and Dentistry, The University of Western Ontario, London, ON N6A 3K7, Canada; abbas.jessani@schulich.uwo.ca; 5School of Medicine and Dentistry, Griffith University, Gold Coast 4222, Australia; santoshkumar.tadakamadla@griffithuni.edu.au (S.K.T.); jyothi.tadakamadla@griffithuni.edu.au (J.T.); 6Menzies Health Institute Queensland, Gold Coast 4222, Australia

**Keywords:** oral health, children, impact, parents, families, Saudi Arabia

## Abstract

The impact of poor oral health may not just be limited to the children themselves but can impact their families. The current study aims to perform psychometric analyses of the Arabic version of the Family Impact Scale and investigate the association of its domains with the oral health status of children. This cross-sectional study was carried out in a sample of 500 parent-child dyads from high schools of Jazan city of the Kingdom of Saudi Arabia. The Arabic version of the Family Impact Scale was subjected to reliability and validity tests. The explanatory variables in the current study are: the oral health status, parents combined income, parents’ education, age and sex of the child. The descriptive analysis was reported using proportions, this was followed by the bivariate and multivariable analyses. About 24.2% of children were reported to have fair, poor, and very poor oral health. A lower frequency of family impact corresponded with better oral health (OH) status of children (*p* < 0.001). The likelihood of parent’s taking time off from work and having financial difficulties was nearly two-times greater if their children had poor oral health. Similarly, interruption in sleep and other normal activities of parents is four times and five times greater, respectively, if the child has poor oral health status. Thus, the poor oral health of school children in the Jazan region of Saudi Arabia is a matter of grave concern as it is observed to be associated with family impacts; particularly affecting the parent’s work, sleep, and other normal family activities.

## 1. Background

Oral health conditions are identified in population studies using clinical measures such as the decayed, missing, and filled teeth (DMFT) index for dental caries, or the Dental Aesthetic Index (DAI) for malocclusion. These indices are useful in reporting the prevalence and severity of the oral health status. However, the clinical reporting is less useful to reflect the end-point or the impact of diseases. The World Health Organization (WHO) has considered the oral health-related quality of life (OHRQoL) construct as an integral part of the Global Oral Health Program and as a fundamental measure of an individual’s oral health and well-being [1]. The OHRQoL has been assessed and appraised in various populations and it presents semi-qualitative participant-based ratings of oral health status involving multidimensional characteristics in comparison to the traditional clinical observations [2].

Currently, it is well-understood that untreated oral health conditions may result in impaired daily activities including poor sleeping and eating [3,4,5,6]. However, the impact of oral health conditions may not just be limited to the children themselves but can impact their families [7,8,9]. For instance, thought of failure to instil good oral health practices in their children leading to the oral diseases may render guilt and distress among the parents. Besides, the visit to a dental health care service provider to seek dental treatment for their children by taking time out of their working hours may affect them financially [10,11].

Earlier, Locker and colleagues [12], in agreement with Rothman and colleagues [13], stated that there are several reasons behind the inclusion of the Family Impact Scale in the health-related quality of life instruments. First, the vital role played by parents and caregivers in the health of children; second, the possibility that the long-term disease condition will affect the parents and caregivers; third, the treatments to illnesses mostly address the family needs and concerns along with the child’s; fourth, the fact that the caregiver- or parent-perceived health of children may be persuaded by the level of physical and emotional influence on them by the health of a child.

Recently, Kumar and colleagues further emphasized the inclusion of family impacts and additionally reported that the caregiver burden in subjective measures can be addressed if the parental reports of the child’s health are assessed alongside [14]. Therefore, the family impacts is an important outcome measure that may be associated with oral health conditions in children [15]. Providing such measures that reflect the consequences of untreated oral diseases may contribute to a paradigm shift towards the impact of oral diseases and the assessment of the biopsychosocial approach to oral health care [2]. This will augment the data on oral health of children in and around a country or region, and help the public health practitioners to advocate the prevention strategies more effectively.

The Jazan region of Saudi Arabia needs effective oral disease prevention strategies. The oral health of school children in this region has been assessed previously and is observed to be rather poor. In a study that sampled 520 boys and 333 girls from the Jazan region, nearly 91% of them had at least one decayed tooth in their oral cavity [16]. These results were consistent with another recent study that utilized self-perceived measure to assess the teeth condition of school-going children [17]. However, the OHRQoL construct and subsequently the Family Impact Scale has not been administered in this region, thus portraying the dearth of imperative evidence. Moreover, within the 25 Arabic speaking nations across the globe, there is no Arabic version of the Family Impact Scale. Thus, the current study aims to perform psychometric analyses of the Arabic version of the Family Impact Scale and investigate the association of its domains with the oral health status of children.

## 2. Methodology

### 2.1. Ethical Considerations

The current study is carried out in accordance with the Declaration of Helsinki and the Human Research and Ethics Committee of the Jazan University, Saudi Arabia approved the study protocol (Approval record: REC42/1/090). Each participant was recruited in the study only after providing signed informed consent.

### 2.2. Study Design, Participants, and Sampling

The current cross-sectional study design comprises two phases. In phase 1, the original English version of the Family Impact Scale was subjected to psychometric analyses. For the cross-cultural adaptation, the current study followed the procedure specified in an earlier report [14]. The original scale was translated into Arabic by a practicing dentist who is familiar with both languages (English and Arabic). Another bilingual dentist was asked to reverse translate the Arabic version to English and the translated documents were presented in a virtual meeting to discuss and sort the discrepancies. The translated Arabic version was then shared with school teachers (*n* = 2) and parents (*n* = 2) to obtain their view on the clarity of each item. It was enquired if the items were confusing, upsetting, and difficult in terms of understanding or wording. There was no difficulty reported, thus, completing the translation procedure. Forty parents, who were not part of the later investigations received the questionnaire twice with a gap of two weeks to compute the test-retest reliability.

In phase 2, the tested questionnaire was distributed to the study participants. The process involved creating an online survey and disseminating the link to the parents of the primary school children of Jazan city, targeting only the Saudi Nationals; this contributed to the homogeneity of the study sample. The link which was created using Google Forms was circulated using email or the WhatsApp messaging service with the help of the school administration staff. The landing page of the survey indicated that completion and submission of the online survey implied the formal consent of the parent.

The sample size calculation for phase 2 was estimated using the values from a prior study, where 80.3% of children were found to have poor oral health [17]. The precision and power of the study was set at 0.05 (type-1 error) and 80%, respectively. After accounting for about 20% of non-respondents, the required sample size was 294. However, nearly 500 parents from eight sex-segregated schools in Jazan city provided their responses and were included in the study. We included more than the calculated sample size to increase the power of the study and also to have a more representative sample.

### 2.3. Study Variables

The impacts on the family were considered as the outcome variable in the current study and the data were collected using the Family Impact Scale (FIS) [12] of the Parental-Caregiver Perceptions Questionnaires (P-CPQ) [18]. The FIS questionnaire includes items on parental/family activity (five questions); parental emotions (four questions); family conflicts (four questions); and financial burden (one question). The explanatory variables included child oral health status as perceived by their parents (responses: very poor, poor, fair, good, very good), parents combined income (≤SAR10,000, >SAR 10,000 per month), parents’ education (did not attend/complete school, completed schooling, completed graduate-level education), age of the child (in years), and sex of child (boy or girl).

### 2.4. Statistical Analysis

The convergent validity of the Arabic version was computed by assessing the correlations (Spearman’s Correlation Coefficient) of subscale and overall scores with global ratings of oral health. A value below 0.20 was regarded as weak, between 0.20 and 0.30 as a medium, and from 0.30 onwards as high [19]. Internal consistency of subscale and overall subscale was computed using Cronbach’s alpha; a value of 0.70 and more was considered acceptable [14]. Intra-class correlation coefficient (ICC) was used to report test-retest reliability and a value of 0.80 was considered as good and more than 0.80 as excellent [20]. The descriptive analysis is reported using proportions, this is followed by computing the unadjusted and adjusted odds ratio estimates. The outcome and explanatory variable i.e., family impacts and oral health status were recategorized into dichotomous before carrying out the binary logistic regression analyses. The final analyses were adjusted for the sociodemographic characteristics. All analyses were carried out using the SPSS version 25.0 (IBM, New York, NY, USA).

## 3. Results

Parents (mother or father) of 500 children (55.6% boys and 44.4% girls) responded to the questionnaire. Nearly a quarter (24.2%) of children were reported to have fair, poor, and very poor oral health. The descriptive findings are presented in Table 1. The findings from the family impact scale indicated that due to poor oral health of children, about 21% of parents had to take time off from their work, 36.8% commented that their children needed more attention, 24.4% had less time for themselves, 37.6% had sleeping problems, and 16.8% had financial difficulties. About 22% of children feel jealous and 16.8% blame parents for their poor oral health.

The findings from discriminant validity tests showed a significant gradient in mean FIS scores across various categories of oral health status (Table 2). The overall scale demonstrated a Cronbach’s alpha value of 0.86, indicating a high internal consistency. Similarly, the subscales also showed moderate to high internal consistency; the financial burden subscale showed excellent internal consistency. The test-retest reliability based on the forty recruited parents indicated a perfect agreement (0.90), and similar were the results across the subscales of FIS (Table 3).

The analysis in Table 4 indicates that a lower frequency of family impact corresponds to better oral health status of children (*p* < 0.001). This finding is consistent with each of the item in the family impact scale. Table 5 presents the association between the oral health status of children and the family impacts after adjusting for confounders. The likelihood of parent’s taking time off from work and having financial difficulties is nearly two-times greater if their children had poor oral health. Similarly, interruption in sleep and other normal activities of parents is four times and five times greater, respectively, if their children have poor oral health.

## 4. Discussion

A child does not live in isolation; poor oral health may not just directly impact them physically, emotionally, and socially; leading to impaired nutrition, communication, school attendance, self-esteem etc., but could also present a more distal affect related to their parents and families. Thus, untreated oral health conditions may have wider repercussions, measuring this impact is particularly relevant, as it will assist the researchers and policymakers in advocating and raising awareness for preventing oral diseases. The current study validates the Arabic version of the Family Impact Scale. The internal consistency and the test-retest reliability was excellent, and the hypothesis relating to the construct validity was successfully affirmed. These findings were consistent with earlier reports from Canada [12], India [14], Italy [21], Saudi Arabia [22], and Brazil [23]. Many of the earlier reports utilized the clinical oral health data which are supposedly a better measure in comparison to self- or parent-perceived measures. However, given the current pandemic situation (COVID-19) and that the schools followed a distant-learning process, it was not feasible to clinically examine the oral health statuses of children. Thus, the link to the questionnaire was emailed or sent through the WhatsApp messaging service. Nevertheless, parent’s perception to determine oral health status of children has been used earlier and found to somewhat represent the clinically examined oral health status [17,24,25]. Findings show that, because of the poor oral health condition of children, the parents reported the need to take time off from work, impaired sleep and regular activities, and financial distress, and this result was similar to the response received from parents residing in a more developed country with higher Gross Domestic Product (GDP) and a better healthcare infrastructure [26].

Nearly a quarter of the children in the current study were reported by their parents to have poor oral health. This is undervalued in comparison to the reports from research carried out in a similar study setting where a greater proportion of children had poor teeth and gum conditions [17,27] and the underestimation can be attributed to the lower oral health literacy among the adults residing in the Jazan region of Saudi Arabia [28]. Findings also show that poor oral health in children is significantly related to all the domains of the family impact scale, implying that the poor oral health status of children significantly impacts the daily life and well-being of their parents. Interestingly, the poor oral health of children was associated with greater odds of financial difficulty to parents, and this is despite free oral health treatment facilities available in the region. Parents also reported guilt and stress because of the oral health status of their children, and this may be due to the disability caused because of pain and infection related to untreated oral health conditions. Most of the findings demonstrated a better precision after adjusting for the socio-economic status of parents. Abed and colleagues demonstrated similar findings using a much larger sample size in a nationwide study carried out in the United Kingdom [26]. 

One of the strengths of the current study was that the sample size was larger than what was required, contributing to adequate power of the study and a significant representation of high school children in the region. Next, a tested assessment tool was used to gather data adding to the reliability and validity of the findings. Lastly, the sample consisted of only Arab nationals which stipulated a homogenous study sample, thus, covering for bias arising due to culturally different oral hygiene practices. However, the current study is cross-sectional and may not indicate a causal association between oral health and family impacts. The selection of sample was non-random and this may affect the generalization of the findings. Nevertheless, the sample participation rate was good, and the current study did not intend to report any population estimates but aimed to investigate the impact of oral health status on the child’s families. Additionally, findings from clinical oral health examinations would have yielded more accurate data on oral health, but this was not feasible given the current pandemic situation. Nonetheless, the subjective assessment of oral health is followed by many research studies measuring the OHRQoL of children, and the findings are somewhat consistent with the clinical reports [29,30,31]. Despite the limitations, the current study yields important findings which may significantly contribute to the lack of literature from Arab populations.

In conclusion, the poor oral health of schoolchildren in the Jazan region is a matter of grave concern. It is associated with family impacts; particularly affecting the parent’s work, sleep, normal family activities, leading to distress and guilt. Thus, attending to oral diseases can have positive effects on children and their families. Further investigations using longitudinal study designs and in different study setting are required to substantiate the findings.

## Figures and Tables

**Table 1 children-08-00586-t001:** Descriptive characteristics of the study sample (*n* = 500).

Study Variables	*n* (%)
Sex	
Girl	278 (55.6%)
Boy	222 (44.4%)
Parent-perceived oral health	
Very poor	7 (1.4%)
Poor	13 (2.6%)
Fair	101 (20.2%)
Good	135 (27%)
Very good	242 (48.4%)
FIS	
Father or Mother took time off from work	
No	396 (79.2%)
Yes	104 (20.8%)
Child needed more attention from parents	
No	316 (63.2%)
Yes	184 (36.8%)
Less self-time for parents	
No	378 (75.6%)
Yes	122 (24.4%)
Sleeping problems for parents	
No	312 (62.4%)
Yes	188 (37.6%)
Interruption to family activities	
No	434 (86.8%)
Yes	66 (13.2%)
Parent’s feeling upset	
No	414 (82.8%)
Yes	86 (17.2%)
Parent’s feeling guilty	
No	393 (78.6%)
Yes	107 (21.4%)
Have fewer opportunities in life	
No	407 (81.4%)
Yes	93 (18.6%)
Feel uncomfortable in public places	
No	380 (76%)
Yes	120 (24)
Child argues	
No	406 (81.2%)
Yes	94 (18.8%)
Child feels jealous	
No	390 (78%)
Yes	110 (22%)
Conflict in family	
No	445 (89%)
Yes	55 (11%)
Child blames parents	
No	429 (85.8%)
Yes	71 (14.2%)
Financial difficulties	
No	416 (83.2%)
Yes	84 (16.8%)

**Table 2 children-08-00586-t002:** Findings from discriminant validity of the overall and subscale scores by perceived oral health status.

Oral Health Status	FIS Overall Score		Parental Activities		Parental Emotions		Family Conflict		Financial Burden	
	Median	Mean (SD)	Median	Mean (SD)	Median	Mean (SD)	Median	Mean (SD)	Median	Mean (SD)
Very poor	7.5	8.0 (7.3)	6.0	5.0 (4.8)	6.0	5.1 (4.0)	7.0	6.0 (4.8)	8.0	8.0 (4.8)
Poor	6.5	7.7 (6.0)	5.0	3.6 (3.3)	4.0	4.6 (3.0)	6.0	5.6 (3.3)	6.0	7.6 (3.3)
Fair	5.0	7.0 (7.6)	4.5	3.1 (3.2)	3.5	3.1 (3.2)	5.5	4.1 (3.2)	5.5	5.1 (3.2)
Good	2.0	3.0 (5.9)	2.0	3.0 (3.9)	2.0	3.9 (3.0)	2.5	2.0 (3.9)	2.0	3.0 (3.9)
Very good	1.5	2.5 (4.1)	1.0	2.5 (3.0)	1.5	2.5 (2.0)	1.5	1.5 (3.0)	1.0	1.5 (3.0)

**Table 3 children-08-00586-t003:** Internal consistency and intra class correlation coefficient of the family impact scale.

FIS	Internal Consistency (Cronbach’s Alpha)	Intraclass Correlation Coefficient (ICC, 95% CI)
Overall scale	0.86	0.90 (0.80, 0.93)
Subscales		
Parental activity	0.87	0.92 (0.68, 0.96)
Parental emotions	0.77	0.76 (0.58, 0.84)
Family conflict	0.80	0.82 (0.58, 0.90)
Financial burden	0.90	0.92 (0.68, 0.94)

**Table 4 children-08-00586-t004:** Distribution of family impacts by the oral health status of children (*n* = 500).

Family Impact Scale	Oral Health Status					*p*-Value
	Very Poor	Poor	Fair	Good	Very Good	
Father or Mother taken time off from work						
No	1 (0.2%)	5 (1%)	57 (11.4%)	108 (21.6%)	223 (44.6%)	<0.001
Yes	6 (1.2%)	8 (1.6%)	44 (8.8%)	27 (5.4%)	19 (3.8%)	
Child needed more attention from parents						
No	3 (0.6%)	3 (0.6%)	46 (9.2%)	85 (17%)	178 (35.6%)	<0.001
Yes	4 (0.8%)	10 (2%)	55 (11%)	50 (10%)	64 (12.8%)	
Less self-time for parents						
No	5 (1%)	8 (1.6%)	64 (12.8%)	96 (19.2%)	204 (40.8%)	<0.001
Yes	2 (0.4%)	5 (1%)	37 (7.4%)	39 (7.8%)	38 (7.6%)	
Sleeping problems for parents						
No	2 (0.4%)	3 (0.6%)	43 (8.6%)	82 (16.4%)	182 (36.4%)	<0.001
Yes	5 (1%)	10 (2%)	58 (11.6%)	53 (10.6%)	60 (12%)	
Interruption to family activities						
No	3 (0.6%)	11 (2.2%)	79 (15.8%)	119 (23.8%)	220 (44%)	<0.001
Yes	4 (0.8%)	2 (0.4%)	22 (4.4%)	16 (3.2%)	22 (4.4%)	
Parent’s feeling upset						
No	5 (1%)	5 (1%)	79 (15.8%)	107 (21.4%)	216 (43.2%)	<0.001
Yes	2 (0.4%)	8 (1.6%)	22 (4.4%)	28 (5.6%)	26 (5.2%)	
Parent’s feeling guilty						
No	2 (0.4%)	7 (1.4%)	60 (12%)	105 (21%)	217 (43.4%)	<0.001
Yes	5 (1%)	6 (1.2%)	41 (8.2%)	30 (6%)	25 (5%)	
Have fewer opportunities in life						
No	5 (1%)	8 (1.6%)	74 (14.8%)	104 (20.8%)	214 (42.8%)	<0.001
Yes	2 (0.4%)	5 (1%)	27 (5.4%)	31 (6.2%)	28 (5.6%)	
Feel uncomfortable in public places						
No	1 (0.2%)	8 (1.6%)	72 (14.4%)	102 (20.4%)	195 (39%)	<0.001
Yes	6 (1.2%)	5 (1%)	29 (5.8%)	33 (6.6%)	47 (9.45%)	
Child argues						
No	3 (0.6%)	8 (1.6%)	74 (14.8%)	102 (20.4%)	217 (43.4%)	<0.001
Yes	4 (0.8%)	5 (1%)	27 (5.4%)	33 (6.6%)	25 (5%)	
Child feels jealous						
No	4 (0.8%)	8 (1.6%)	69 (13.8%)	97 (19.4%)	210 (42.5%)	<0.001

**Table 5 children-08-00586-t005:** Logistic regression models demonstrating the impact of poor oral health status of children on selected items in the family impact scale (*n* = 500).

Oral Health Status	Unadjusted OR (95%CI)	*p*-Value	* Adjusted OR (95%CI)	*p*-Value
Time off from work				
Good	1.00		1.00	
Poor	1.9 (1.02, 2.72)	<0.001	2.1 (1.16, 2.54)	<0.001
Financial difficulties				
Good	1.00		1.00	
Poor	2.3 (1.45, 3.62)	<0.001	1.9 (1.49, 2.51)	<0.001
Child need more attention				
Good	1.00		1.00	
Poor	4.1 (2.27, 7.26)	<0.001	4.0 (2.15, 7.75)	<0.001
Parent’s sleep disturbed				
Good	1.00		1.00	<0.001
Poor	5.1 (2.73, 9.50)	<0.001	4.9 (2.80, 8.76)	
Interruption of normal activities				
Good	1.00		1.00	
Poor	4.5 (2.30, 9.01)	<0.001	3.1 (2.78, 6.76)	<0.001
Parent’s feel guilty				
Good	1.00		1.00	
Poor	5.4 (2.90, 9.10)	<0.001	6.4 (3.30, 9.26)	<0.001

Good oral health status is summarized by including the responses good and very good. Poor oral health status is summarized by including the responses fair, poor, and very poor. * OR analyses are adjusted for parent’s education and income.

## Data Availability

The data is available upon reasonable request.

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
