# Peer review of "Impact of the Poor Oral Health Status of Children on Their Families: An Analytical Cross-Sectional Study"

_children, 2021, doi:10.3390/children8070586_

Round 1
Reviewer 1 Report
The manuscript submitted for review, entitled „Impact of the poor oral health status of children on their families: an analytical cross-sectional study” is the result of socio-medical research.
Interesting research results, generally not new, confirmed by data from the literature, but new for a selected large population of children- parents from Jazan city of the Kingdom of Saudi Arabia.
The Authors saw the necessity to conduct a study of the children's oral health, however, during the pandemic, they decided to conduct a survey, which was very well prepared.
Of course, the most objective would be a clinical dental examination by dentists, which the authors point to in the manuscript.
The test results are presented in 5 tables. I would pay attention to table 1 – if % is marked in the header of the table, there is no need to write next to each result.
Generally, when quoting, it is assumed that, for example, Smith et all.
Line 198: „Nearly, a quarter of the children in the current study were reported to have poor oral health” - whether it's data from a questionnaire that parents have completed or from previous clinical trials. It is unclear.
It would be good to develop the discussion by presenting more closely the results of similar studies conducted in other populations - but this is only a suggestion - in my opinion, this would enrich the manuscript and show the differences between other child populations.
Conclusions are well structured and respond to the goals of the work.
Author Response
Reviewer 1:
The manuscript submitted for review, entitled Impact of the poor oral health status of children on their families: an analytical cross-sectional study” is the result of socio-medical research.
Interesting research results, generally not new, confirmed by data from the literature, but new for a selected large population of children- parents from Jazan city of the Kingdom of Saudi Arabia.
The Authors saw the necessity to conduct a study of the children's oral health, however, during the pandemic, they decided to conduct a survey, which was very well prepared.
Of course, the most objective would be a clinical dental examination by dentists, which the authors point to in the manuscript.
Response: Thank you.
The test results are presented in 5 tables. I would pay attention to table 1 – if % is marked in the header of the table, there is no need to write next to each result.
Response: We agreed to the suggestion and have now removed the % signs from Table 1.
Generally, when quoting, it is assumed that, for example, Smith et all.
Line 198: „Nearly, a quarter of the children in the current study were reported to have poor oral health” - whether it's data from a questionnaire that parents have completed or from previous clinical trials. It is unclear.
Response: It is from the questionnaire that the parents have completed. We have now changed the sentence to sound more meaningful.
It would be good to develop the discussion by presenting more closely the results of similar studies conducted in other populations - but this is only a suggestion - in my opinion, this would enrich the manuscript and show the differences between other child populations.
Response: We have now added more substance to the discussion. This could be observed at the first paragraph and also while discussing the limitation of the study.
Conclusions are well structured and respond to the goals of the work.
Response: Thank you.
Reviewer 2 Report
The authors did a good job presenting the impacts of children's oral health status on the family. As mentioned by the authors, there is a lack of clinical data on the children's oral health condition, which limits the scope of the manuscript to some extent, however, still presents a great deal of information and insight into the topic of interest. I would advise the authors to follow the suggestions.

Author Response
Reviewer 2
Thank you.
All the comments have been accepted and the changes are reported in the current version of the manuscript.
Round 2
Reviewer 2 Report
The authors have made all the necessary changes. I have suggested one minor change to be made in the descriptive table.
